# Spectral Dependence of the Energy Transfer from Photosynthetic Complexes to Monolayer Graphene

**DOI:** 10.3390/ijms23073493

**Published:** 2022-03-23

**Authors:** Marcin Szalkowski, Alessandro Surrente, Kamil Wiwatowski, Zhuo Yang, Nan Zhang, Julian D. Janna Olmos, Joanna Kargul, Paulina Plochocka, Sebastian Maćkowski

**Affiliations:** 1Faculty of Physics, Astronomy and Informatics, Institute of Physics, Nicolaus Copernicus University, 87-100 Torun, Poland; m.szalkowski@intibs.pl (M.S.); wiwatowski@umk.pl (K.W.); 2Institute of Low Temperature and Structure Research, Polish Academy of Sciences, ul. Okolna 2, 50-422 Wroclaw, Poland; 3Laboratoire National des Champs Magnétiques Intenses, EMFL, CNRS UPR 3228, 31400 Toulouse, France; alessandro.surrente@pwr.edu.pl (A.S.); zhuo.yang@issp.u-tokyo.ac.jp (Z.Y.); swznlz@163.com (N.Z.); paulina.plochocka@lncmi.cnrs.fr (P.P.); 4Department of Experimental Physics, Faculty of Fundamental Problems of Technology, Wroclaw University of Science and Technology, 50-370 Wroclaw, Poland; 5Solar Fuels Laboratory, Center of New Technologies, Banacha 2C, University of Warsaw, 02-097 Warsaw, Poland; julian.janna.olmos@uj.edu.pl (J.D.J.O.); j.kargul@cent.uw.edu.pl (J.K.)

**Keywords:** graphene, energy transfer, fluorescence, photosynthetic complexes

## Abstract

Fluorescence excitation spectroscopy at cryogenic temperatures carried out on hybrid assemblies composed of photosynthetic complexes deposited on a monolayer graphene revealed that the efficiency of energy transfer to graphene strongly depended on the excitation wavelength. The efficiency of this energy transfer was greatly enhanced in the blue-green spectral region. We observed clear resonance-like behavior for both a simple light-harvesting antenna containing only two chlorophyll molecules (PCP) and a large photochemically active reaction center associated with the light-harvesting antenna (PSI–LHCI), which pointed towards the general character of this effect.

## 1. Introduction

Graphene is a two-dimensional hexagonal lattice of carbon atoms in sp2 hybridization. This structure yields unique electrical or thermal properties, which are a direct consequence of the energy band structure, i.e., linear dispersion relation in the vicinity of Dirac points [1,2], in addition to high carrier mobility and thermal stability, which opens potential for applications of graphene in electronics. The absence of the energy gap in this material translates into nearly constant absorption of light over the whole visible spectral range. Indeed, a single atomically thin layer of graphene [1] absorbs 2.3% of incoming light, regardless of the wavelength. At the same time graphene is not fluorescent. Thus, all captured energy is dissipated in a nonradiative way, presumably via carrier– carrier and carrier–phonon scattering. These properties make graphene an almost ideal energy acceptor in hybrid assemblies, and thus, it can be used to examine energy transfer mechanisms and processes that take place at the nanoscale.

Energy transfer in hybrid nanostructures that include graphene as the energy/charge acceptor has been studied for a variety of emitters, including molecules, polymers, and semiconductor nanocrystals, both in solution and in a layer geometry [3,4]. The proximity of graphene strongly influences the optical properties of emitters. First of all, the fluorescence intensity for emitters is strongly quenched. This quenching, which can be as high as 99%, is accompanied with dramatic shortening of fluorescence lifetimes, indicating efficient energy transfer to graphene in such assemblies. Moreover, the two-dimensional character of graphene results in a weaker dependence of the energy transfer efficiency on the distance between a fluorophore and graphene, as compared with the case of two interacting dipole moments, where the efficiency of the Forster resonance energy transfer (FRET) is inversely proportional to the sixth power of the distance. As discussed previously, while graphene cannot be considered as a classical dipole moment, the efficiency of the energy transfer scales with the fourth power of the distance [5,6,7]. Such a dependence has also been observed experimentally [3,8].

Recently, research has focused on the energy transfer in hybrid graphene assemblies has been extended towards systems that are more complex than molecules or nanocrystals, such as the natural pigment–protein complexes that take part in photosynthesis [6,7]. Such hybrid assemblies can be considered one of the pathways aimed at obtaining a “green” device for efficient energy conversion [9]. Indeed, a biomimetic solar cell, in which the photoactive medium is composed of photosynthetic reaction centers, could be a promising solution for conversion and storage of the sunlight energy [10,11,12,13,14,15,16,17,18,19,20,21,22,23,24,25,26,27,28]. This concept is based on the fact that photosynthetic complexes, being evolutionarily optimized for over three billion years, are highly efficient photochemical devices. Because of the specific arrangement of various cofactors present within the photosynthetic reaction center, light absorption, energy transfer, and photochemistry are optimized, so that every photon captured by the reaction center is converted into a charge-separated state that is required for electron transfer and, ultimately, the production of a proton gradient across the thylakoid membrane and initiation of synthesis of carbon-based high energy molecules [9]. Interfacing photosynthetic proteins as building blocks of solar energy conversion devices with other nanostructures has been widely discussed in literature, especially in the context of devising efficient solar-driven charge separation or water splitting [9,21,24,25,29]. These include classic electrode materials such as gold [30], indium tin oxide (ITO) [31,32,33], and TiO2 [34,35,36], but also graphene and its derivatives [15,17,18,19,22,28,37,38]. Another rather promising route has been interfacing photosynthetic pigment–protein complexes with metallic nanostructures in order to couple the excited states of the inbound pigments with localized surface plasmon resonances [11,18,39,40,41,42,43,44,45]. In this way, the optical absorption of photosynthetic complexes can be increased, typically by over an order of magnitude.

An important issue that needs to be addressed when photosynthetic proteins are deposited on various nanostructures relates to the modifications of the optical properties of such hybrid systems due to mutual interactions between the components. Among the most surprising effects encountered for relatively simple light-harvesting antennae, such as peridinin-chlorophyll *a*-protein (PCP), is that fluorescence decay times measured for PCP deposited on graphene change with the excitation wavelength. Specifically, it was shown that upon excitation with 405 nm, the decay times were substantially shorter than those obtained with 630 nm excitation [7]. This result, based on just two excitation wavelengths, was in clear contrast with a classical description of the energy transfer, where the fluorescence decay time, which is a measure of the efficiency of energy transfer [5,46,47], is independent of the excitation wavelength. Such an effect would have important repercussions for any architecture that would attempt to utilize graphene as a component of biomimetic solar energy conversion devices.

In this work, we aimed at precise determination of the wavelength dependence of the energy transfer efficiency in a hybrid structure in which photosynthetic complexes of various size, composition, and function were deposited on a monolayer graphene. First, we designed and performed a systematic fluorescence excitation microscopy analysis, where the emission intensity was monitored for varying excitation wavelengths. Second, we carried out this analysis both for the PCP complex and for a photosystem I with its associated light-harvesting antenna (PSI–LHCI), that has been previously used in various types of biohybrid solar energy conversion devices [12,13,17,18,19,23,28,48]. Importantly, both pigment–protein complexes studied in this work, despite chlorophyll-based emission, differed significantly in the structure and function of the photosynthetic apparatus. The PCP complexes were small and relatively simple antenna complexes optimized for efficient light harvesting and transfer of the captured energy toward the photosynthetic reaction center. On the other hand, the PSI–LHCI was a photosynthetic reaction center that was considerably larger than PCP and much more complex in pigment and protein assembly. It combined multiple roles, such as direct light harvesting or energy capturing from the outer antenna complexes and transferring this energy to the photochemical reaction center—a special P700 chlorophyll *a* pair, where charge separation occurred.

The results of fluorescence excitation microscopy measurements indicated that for both photosynthetic proteins (PCP and PSI–LHCI), a substantial reduction in fluorescence intensity occurred upon excitation into the blue-green spectral region. Importantly, the spectral dependence of the emission intensity exhibited a resonance-type character. These results point toward a more complex role of graphene in assemblies in which both energy transfer and tightly focused illumination take place.

## 2. Results

### 2.1. Spectroscopic Characterization of Materials

In Figure 1, we show optical spectra measured for both PCP (Figure 1a) and PSI–LHCI (Figure 1b) in solution. Contributions of both types of pigments, Pers and Chl *a*, were visible in the absorption spectrum of PCP (black line in Figure 1a). Bands around 440 nm and 668 nm corresponded to Chl *a* absorption at the Soret band and Qy band, respectively. A prominent band in the range from 350 nm to 550 nm corresponded to Per absorption. On the other hand, the emission was associated solely with Chl *a* fluorescence, centered at 673 nm (red curve). This, along with close resemblance between absorption and excitation spectra, indicated highly efficient energy transfer between Pers and Chls, reaching 90% [49,50,51,52].

Compared to PCP, the PSI–LHCI complex was much larger, containing nearly 200 cofactors enclosed within the protein scaffold [53,54,55]. The absorption spectrum of this complex (Figure 1b) underlined the dominant role of Chl *a* in light-harvesting; however contributions from Cars were also clearly visible, mostly in the green spectral range. In the case of PSI–LHCI complex, the Chl *a* molecules were responsible for fluorescence emission (red curve), although there were different subpopulations of these fluorophores present in the spectra, as suggested by significant broadening of the emission spectrum. Energy pathways within the PSI–LHCI complex and associated kinetics have been studied previously [40,53,56].

The optical properties of graphene are unique in the sense that it features flat absorption of light across the visible spectral range, and thus, it can be considered as an ideal energy absorber. Our recent investigation of energy coupling between PCP complexes and graphene indicated that the fluorescence decay time depended on the excitation wavelength [7]. This observation suggests that the efficiency of the energy transfer depends on the excitation wavelength.

### 2.2. Excitation Wavelength Dependence of the PCP Emission

In order to systematically study the excitation wavelength dependence on the emission properties of photosynthetic complexes deposited on a graphene monolayer, we conducted a photoluminescence excitation experiment using a microscope objective for both excitation and light collection. By tuning the excitation wavelength, we measured fluorescence spectra and extracted emission intensities. There are two important comments needed before the discussion of the results. First of all, it was necessary to perform such an experiment in a microscopic setup, as we wanted to probe only these complexes that were in the vicinity of graphene. This imposed another criterion that needs to be fulfilled, namely, the very small thickness of the PVA layer containing the photosynthetic complexes. On the other hand, pigment–protein complexes are highly suitable for studying energy transfer processes, as the protein scaffold itself provides a spacer between the pigments and graphene, which, while not very thick, is sufficient to partially inhibit the energy transfer. In other words, should the pigment molecules be placed directly onto graphene, the efficiency of the energy transfer would be extremely, high leading to essentially complete quenching of fluorescence emission, regardless of the experimental conditions [3,5,57].

Figure 2 shows a sequence of fluorescence spectra collected for PCP complexes at different excitation wavelengths. We tuned the excitation wavelengths from 610 to 400 nm with a step size of 5 nm while keeping the laser spot in the same location. In this way, we minimized any effects related to fluctuations of either PCP concentration or PCP layer thickness between different points. In this context, keeping the sample at cryogenic temperatures was highly important, as it considerably reduced any photobleaching of pigments. The fluorescence spectra measured for PCP deposited on glass (black) or graphene (red) featured identical position, shape, and line width, which indicated that the protein was intact, and all its functionality was maintained, for both types of assemblies. However, the wavelength dependence differed considerably between the PCP complexes on graphene and on glass. First of all, the intensities measured for PCP on graphene were lower than those measured for the reference sample, which fact may be attributed to the energy transfer between both components of the assembly [3,4,6,7,58]. The direct comparison of actual emission intensities would have been questionable, as we did not know the precise concentration of PCP complexes at a given position of the assembly. However, the variation of relative intensities was much more relevant. Specifically, while for the PCP complexes on glass, the highest emission intensity was observed for the excitation wavelength of 480 nm, the intensity measured for the same excitation wavelength for PCP on graphene was much smaller. This observation suggests that the efficiency of the energy transfer from PCP to graphene indeed depended on the excitation wavelength applied [7]. Qualitatively, a similar result was obtained for the PSI–LHCI complexes.

### 2.3. Excitation Wavelength Dependence of Emission Intensity of PCP and PSI–LHCI

The emission intensities extracted by integrating the spectra on a graphene monolayer and on glass for PCP and PSI–LHCI complexes are compared in Figure 3a,b, respectively. The shape of the dependence obtained for PCP on glass was essentially the same as the absorption and excitation spectra measured for PCP solution (see Figure 1). This result implied that the PCP complexes were functional upon fabrication of the layer and during the measurement and that the optical properties of the PCP complexes were not influenced by any of manipulations during the assembly preparation. In particular, we observed no effect of embedding the pigment–protein complexes in the PVA polymer layer on the optical properties.

In contrast, the excitation spectrum of PCP complexes on graphene was radically different, in that a substantial and systematic decrease in emission intensity was observed for the blue-green spectral range. The emission intensity seemed to recover for the wavelengths shorter than 400 nm. The experiment was repeated for more than 10 spots on each sample, and the result was always the same as shown in Figure 3a. Therefore, we concluded that for emitters deposited on a monolayer graphene, the efficiency of the energy transfer depended on the excitation wavelength.

Similar experiments performed for PSI–LHCI on graphene yielded qualitatively identical results (see Figure 3b). This indicated that the effect observed for a relatively simple light-harvesting pigment–protein complex, such as PCP, was not exclusive to this complex but had a more universal character. Indeed, for large light harvesting/charge-separating pigment–protein complexes, such as the PSI–LHCI complex, the energy transfer to graphene was also wavelength–dependent. This finding is important and should be considered for any hybrid architecture aimed at exploiting graphene as a component of energy conversion devices or sensors.

## 3. Discussion

By dividing the excitation spectra measured for photosynthetic complexes deposited on graphene by the corresponding reference, we gained insight into the character of this wavelength-dependent interaction between both modules of the hybrid assembly. As shown in Figure 4, in both cases, we found a resonance–like behavior. The linewidths of these resonances were similar for the two photosynthetic complexes studied in this work, while the maxima were slightly shifted. The shapes of these curves were rather similar to plasmon resonances in metallic nanoparticles, suggesting that the behavior observed for PCP and PSI–LHCI complexes deposited on graphene could be associated with the presence of free electrons in graphene. However, more research should be conducted in order to elucidate the underlying mechanism of this unique effect in greater detail.

## 4. Materials and Methods

Graphene substrates were purchased from Graphene Supermarket^®^ (Ronkonkoma, NY, USA). For both types of photosynthetic pigment–protein complexes, we used single-layer graphene (SLG) deposited on 1 cm × 1 cm p-doped silicon (p-Si) substrates covered with a 285 nm thick silicon dioxide layer. In the case of SLG, 97% of the surface was a monolayer graphene. The presence of graphene on the p-Si substrate was confirmed by Raman spectroscopy (not shown).

For the optical experiments, we used two photosynthetic complexes: a water-soluble PCP light-harvesting antenna present in *Dinoflagellates* and a PSI–LHCI complex from a red microalga, *Cyanidioschyzon merolae*. Aqueous solution of PCP from *Amphidinium carterae* was purchased from BD Pharmingen (San Diego, CA, USA). The PCP complexes contained only 10 pigments: 8 peridinins (Pers) and 2 chlorophyll (Chl) *a* molecules [43,49,50,59,60,61]. While absorption of PCP is rather broad, ranging from 400 to 660 nm, the fluorescence is associated with the Chls *a* because of very efficient energy transfer from Pers to Chl *a*. The PSI–LHCI complexes from *C. merolae* contained 157–159 Chl *a* molecules [53,54] and 35 carotenoids (Cars) [53,54,62] and were obtained using a protocol including *C. merolae* culture growth, thylakoid isolation, solubilization, and PSI–LHCI complex purification, described in detail previously [19,55]. At the initial steps of the procedure, the crude PSI–LHCI fraction was eluted from the DEAE TOYOPEARL 650 M column, then loaded with solubilized thylakoids with 0.09 M NaCl and applied onto the DEAE TOYOPEARL 650 S column. Next, the fraction of pure PSI–LHCI was eluted with a continuous 0–0.2 M NaCl gradient in the carrier buffer [55]. After that, the PSI–LHCI sample was concentrated to 1 mg/mL Chl a and further purified on the desalting Superdex G-25 column in buffer (40 mM HEPES-NaOH, pH 8, 3 mM CaCl2, 25% (*w*/*v*) glycerol, 0.03% (*w*/*v*) DDM). This was followed by an anion exchange chromatography step using a UNOTM Q12 column [55]. Finally, the solution of pure PSI–LHCI supercomplexes was collected and concentrated to 2–5 mg/mL Chl *a*, snap-frozen in liquid N2, and stored at −80 °C prior to use. The absorption of the PSI–LHCI complex is very broad; thus, similar ranges of excitation wavelengths were applied for both samples.

The samples were prepared as follows: the stock solution of PCP complexes was further diluted to the concentration of 0.05 mg/mL with 0.2% polyvinyl alcohol (PVA). The solution was then spin-coated on the graphene substrate (30 µL at 17 rps). The PSI–LHCI complexes (0.02 mg/mL) were suspended in a buffer composed of 40 mM HEPES, 3 mM CaCl2, and 0.03% DDM, with pH 8.0; resuspended in 0.2% PVA; then spin-coated as above. The reference samples consisted of the same concentration for each pigment–protein complex that was spin-coated onto a glass coverslip.

Absorption, excitation, and emission spectra in solution were measured using a Varian Cary 50 spectrophotometer and Horiba Jobin Yvon Fluorolog 3 spectrofluorometer (Kyoto, Japan), respectively. The optical properties of the hybrid nanostructures composed of pigment–protein complexes embedded in a PVA matrix on a monolayer graphene were studied using photoluminescence excitation (PLE) microscopy by measuring sequences of emission spectra for varied excitation wavelengths. The experiment was performed in a back-scattering geometry. For excitation, the frequency-doubled output of an optical parametric oscillator (OPO), synchronously pumped by a mode-locked Ti:sapphire laser or the second harmonic (SH) of the mode-locked Ti:sapphire laser, was used. The typical temporal pulse width was 140 fs, with a repetition rate of 80 MHz. The laser beam was focused on the sample surface by Mitutoyo M PlanApo 50 × objective (NA 0.55) (Aurora, IL, USA). To avoid photobleaching, the samples were placed in a liquid helium cryostat at 5 K during the entire measurement. In order to filter out the emission of photo-synthetic complexes, we used a set of suitable optical filters (488 LP, 532 LP, and 633 LP). The emission was detected using a CCD camera coupled to an Acton SP2500 monochromator (Princeton Instruments, Acton, MA, USA) equipped with a grating with 600 lines/mm and blaze at 700 nm.

## 5. Conclusions

In this work, we systematically studied the wavelength dependence of the energy transfer between two types of photosynthetic complexes and graphene. The results of microscopic fluorescence excitation spectroscopy indicated that in the blue-green spectral region, the efficiency of the energy transfer was substantially increased compared with that in other wavelengths. This phenomenon, which was proven for both simple light-harvesting pigment–protein complexes and photochemically active reaction centers, opens the unexplored field of tuning the properties of hybrid graphene nanostructures for optimized optoelectronics and photovoltaics applications.

## Figures and Tables

**Figure 1 ijms-23-03493-f001:**
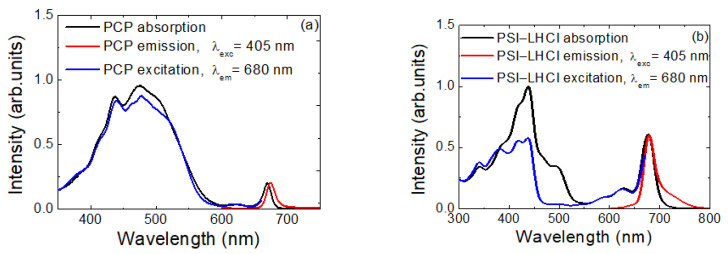
Spectral characteristics of PCP (**a**) and PSI–LHCI (**b**) aqueous-solution absorption spectra (black lines), emission spectra (red lines), and excitation spectra (blue lines).

**Figure 2 ijms-23-03493-f002:**
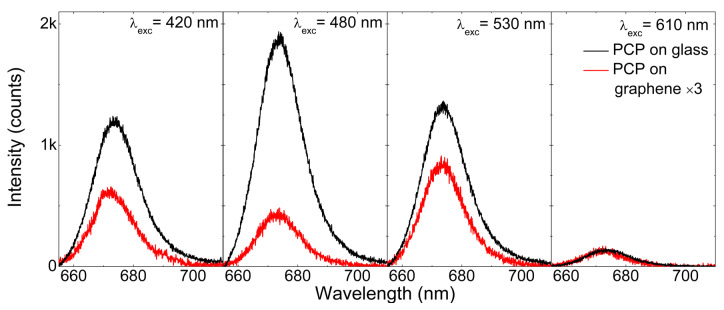
Emission spectra of PCP deposited on glass (black lines) and on graphene substrate (red lines) when excited at different excitation wavelengths. For better comparison, results collected on graphene were multiplied by a factor of 3.

**Figure 3 ijms-23-03493-f003:**
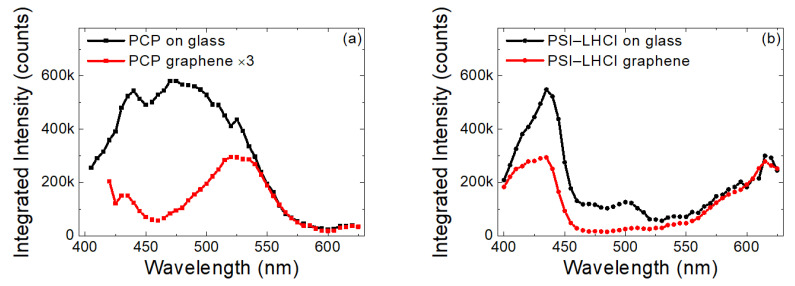
Photoluminescence excitation spectra of PCP (**a**) and PSI–LHCI (**b**) layers deposited on glass (black) and graphene substrate (red).

**Figure 4 ijms-23-03493-f004:**
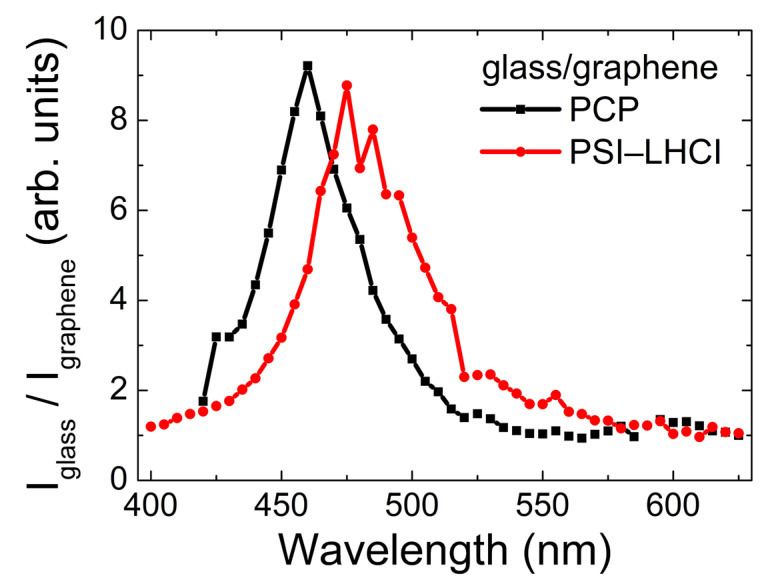
Wavelength dependence of pigment–protein complexes’ fluorescence intensity ratios when deposited on glass or graphene substrate, measured for PCP and PSI–LHCI (black and red lines, respectively).

## Data Availability

The data that support the findings of this study are available from the corresponding author upon reasonable request.

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
