# Peer review of "Spectral Dependence of the Energy Transfer from Photosynthetic Complexes to Monolayer Graphene"

_ijms, 2022, doi:10.3390/ijms23073493_

Round 1
Reviewer 1 Report
Comments of MS IJMS - 1638207
The article presents experimental results for the energy transfer from complexes of the photosynthetic apparatus to monolayer graphene. In this form, the publication cannot be published. The publication lacks an in-depth analysis of the observed effects.
Comments:
The section "Results and Discussion" needs to be divided.
In the introduction to describe the role of the used pigment-protein complexes in the photosynthetic apparatus
In materials and methods to give details about the isolation of the used pigment-protein complexes
The authors use the term protein fluorescence, but the term pigment-protein complexes is more appropriate because the fluorescence they study is from chlorophyll.
The description given in materials and methods of pigment protein complexes is better to give in the results.
Figure 1 – the label on the y-axis is not correct. The figure shows both the absorption and emission and excitation spectra.
It is good to give the original spectra on the basis of which Fig.4 is made.
Author Response
The article presents experimental results for the energy transfer from complexes of the photosynthetic apparatus to monolayer graphene. In this form, the publication cannot be published. The publication lacks an in-depth analysis of the observed effects.
Comments:
- The section "Results and Discussion" needs to be divided.
The section has been divided in the revised version of the manuscript.
- In the introduction to describe the role of the used pigment-protein complexes in the photosynthetic apparatus
The information about the roles of the pigment-protein complexes (PCP and PSI-LHCI) used in the experiment was added to the introduction of the revised version of the manuscript.
- In materials and methods to give details about the isolation of the used pigment-protein complexes
In the revised version of the manuscript the isolation procedure of the PSI-LHCI complexes is included. The solution of PCP complexes is commercially available.
- The authors use the term protein fluorescence, but the term pigment-protein complexes is more appropriate because the fluorescence they study is from chlorophyll.
Thank you for this comment, the term “pigment-protein complex” is more precise to describe the structures under investigation. We corrected this point this in the revised version of the manuscript.
- The description given in materials and methods of pigment protein complexes is better to give in the results.
We believe that this information fits better into the Materials and methods section, were detailed procedure of PSI-LHCI purification is presented along with the structures. The major focus of this work is spectroscopic studies of interactions with graphene and the present structure of the manuscript reflects this fact.
- Figure 1 – the label on the y-axis is not correct. The figure shows both the absorption and emission and excitation spectra.
Thank you for this comment. We changed the y-axis label to “Intensity” in the revised version of the manuscript.
- It is good to give the original spectra on the basis of which Fig.4 is made.
In fact, the spectra used to obtain Fig. 4 are presented in Fig. 3, where broad band excitation spectra of both pigment-protein complexes under investigation are shown for both substrates – bare glass and graphene. The curves presented in Fig. 4 were obtained by dividing the excitation spectra measured for the glass substrate by the ones collected for the graphene substrate.
Reviewer 2 Report
The work of Szalkowski and colleagues presents a short experimental study of the energy transfer from light-absrobing complexes to the graphene monolayer.
The authors show that the wavelength dependence of the energy transfer has a resonance-like character with a maximum in the visible spectral range.
The experiment is well described, the results are quite convincing.
The authors compare the observed phenomenon to plasmonic resonance.
The physical explanation and discussion of the mechanism is absent in the paper.
Overall quality and scientific significance of the work is slightly above average, the text may need minor polishing.
I suggest publishing it.
Author Response
We wish to thank the Reviewer for positive evaluation of our work.
Round 2
Reviewer 1 Report
The authors have taken into account the remarks made by me.